# Functional Analysis of Two Affinity cAMP Phosphodiesterases in the Nematode-Trapping Fungus *Arthrobotrys oligospora*

**DOI:** 10.3390/pathogens11040405

**Published:** 2022-03-26

**Authors:** Ni Ma, Ke-Xin Jiang, Na Bai, Dong-Ni Li, Ke-Qin Zhang, Jin-Kui Yang

**Affiliations:** 1State Key Laboratory for Conservation and Utilization of Bio-Resources, Key Laboratory for Microbial Resources of the Ministry of Education, School of Life Sciences, Yunnan University, Kunming 650091, China; mani916@163.com (N.M.); kxjiang@mail.ynu.edu.cn (K.-X.J.); baina@mail.ynu.edu.cn (N.B.); 18878552280@163.com (D.-N.L.); kqzhang1@ynu.edu.cn (K.-Q.Z.); 2Yunnan Center for Disease Control and Prevention, Kunming 650022, China

**Keywords:** *Arthrobotrys oligospora*, phosphodiesterase, conidiation, stress response, trap formation, secondary metabolism

## Abstract

Phosphodiesterases are essential regulators of cyclic nucleotide signaling with diverse physiological functions. Two phosphodiesterases, PdeH and PdeL, have been identified from yeast and filamentous fungi. Here, the orthologs of PdeH and PdeL were characterized in a typical nematode-trapping fungus *Arthrobotrys oligospora* by gene disruption and phenotypic comparison. Deletion of *AopdeH* caused serious defects in mycelial growth, conidiation, stress response, trap formation, and nematicidal efficiency compared to the wild-type strain. In contrast, these phenotypes have no significant difference in the absence of *AopdeL*. In addition, deletion of *AopdeH* and *AopdeL* resulted in a remarkable increase in cAMP level during vegetative growth and trap formation, and the number of autophagosomes was decreased in Δ*AopdeH* and Δ*AopdeL* mutants, whereas their volumes considerably increased. Moreover, metabolomic analyses revealed that many metabolites were downregulated in Δ*AopdeH* mutant compared to their expression in the wild-type strain. Our results indicate that AoPdeH plays a crucial role in mycelial growth, conidiation, stress response, secondary metabolism, and trap formation. In contrast, AoPdeL only plays a minor role in hyphal and conidial morphology, autophagy, and trap formation in *A. oligospora*. This work expands the roles of phosphodiesterases and deepens the understanding of the regulation of trap formation in nematode-trapping fungi.

## 1. Introduction

Fungi can sense the changes in various physical and chemical stimuli in the environment and regulate the expression of intracellular related downstream genes to respond to the extracellular stimuli; this biological process involves many signal transduction pathways, in which heterotrimeric G protein (G protein) signaling is the most conserved signal transduction mechanism in eukaryotes [1,2]. In most eukaryotes, G protein signaling is involved in the transduction of intracellular downstream signaling components by using cyclic adenosine monophosphate (cAMP) as a second messenger to mediate extracellular stimuli [3]. The components of the cAMP signaling cascade are highly conserved signaling modules in various fungi that regulate a series of essential cellular processes in growth, development, and morphogenesis [4,5]. For example, cAMP regulates nutrient induction, pseudohyphal differentiation, cell cycle progression, and stress signaling in yeasts [6,7]; cAMP regulates morphogenesis, cell polarity, and asexual development in *Neurospora crassa* [8] and *Aspergillus fumigatus* [9]; and cAMP also controls the dimorphic transition in addition to virulence in *Ustilago maydis* [10]. The balance of intracellular cAMP levels depends on biosynthesis by adenylyl cyclases and hydrolysis by cAMP phosphodiesterases (PDEs) [11]. The adenylate cyclases convert ATP to cAMP, and the cAMP can bind to the regulatory subunit of protein kinase A (PKA) to release catalytic subunits of PKA, which activate downstream transcription factors and other effectors involved in various biological reactions [12,13]. In contrast, the PDEs can hydrolyze cAMP and regulate the total intensity of the signal cascade by inactivation of cAMP [14].

In fungi, PDEs were first discovered in *Saccharomyces cerevisiae*, containing low-affinity PDE (Pde1/PdeL) and high-affinity PDE (Pde2/PdeH) [15,16]. Pde1 regulates cAMP levels induced by glucose stimulation or intracellular acidification [17,18]. Pde2, in addition to protecting yeast cells from extracellular cAMP, also regulates the basic or steady-state levels of cAMP [7,19]. In addition, the function of PDEs has also been reported in several filamentous fungi. In *Magnaporthe oryzae*, loss of *pdeH* leads to an increase in intracellular cAMP accumulation during nutrient and infectious growth, increased conidial production, and highly reduced pathogenicity. In contrast, there was no significant phenotypic difference between the Δ*pdeL* mutant and the WT [20]. Similarly, loss of *bcpde2* caused severe vegetative hyphal growth defects, decreased conidia production, and affected spore germination and virulence in *Botrytis cinerea*, but the *bcpde1*-deficient strain showed a wild-type (WT) phenotype [13]. Therefore, PDEs play an indispensable role in asexual and pathogenic development in fungi.

Nematode-trapping (NT) fungi are a special group broadly distributed in terrestrial and aquatic ecosystems in the world [21]. NT fungi can produce specific trapping devices (traps) such as adhesive networks, adhesive knobs, and constricting rings to capture nematodes and extract nutrients from their nematode prey [22,23]. *Arthrobotrys oligospora* is a representative species of NT fungi, which can develop adhesive networks for nematode predation [24]. Recently, G protein and related signaling pathways have been proved to be involved in the mycelial development and trap formation of *A. oligospora*, such as receptors of G protein signaling [25,26] and mitogen-activated protein kinase [27,28]. However, the function of PDEs is unknown in *A. oligospora* and other NT fungi. In this study, the role of two orthologous PDEs (AoPdeH and AoPdeL) in the growth, development, and pathogenicity was identified by gene disruption and multi-phenotypic changes between the *A. oligospora* and mutant strains. In addition, the role of AoPdeH in secondary metabolism was investigated by the metabolomic approach.

## 2. Results

### 2.1. Properties and Conserved Domains of PDEs in A. oligospora

Two orthologous PDEs, AoPdeH and AoPdeL, were retrieved from the fungus *A. oligospora*. AoPdeH contains 766 amino acid residues with a molecular mass of 85.38 kDa and an isoelectric point of 6.37; AoPdeL contains 992 amino acid residues with a molecular mass of 112.21 kDa and an isoelectric point of 7.17. Both AoPdeH and AoPdeL contain multiple conserved domains, the AoPdeH contains a 3′5′-cyclic nucleotide phosphodiesterase superfamily domain (IPR036971), a phosphodiesterase domain (IPR003607), a 3′5′-cyclic nucleotide phosphodiesterase catalytic domain (IPR002073), and 3ʹ5ʹ-cyclic nucleotide phosphodiesterase conserved site (IPR023174); and AoPdeL contains a ribonuclease Z/hydroxyacyl glutathione hydrolase-like (IPR036866) and a cAMP phosphodiesterase domain (IPR000396). The AoPdeH shares a high degree of similarity with orthologs from NT fungi *Duddingtonia flagrans* (89.9%) and *Dactylellina haptotyla* (89.8), and it shares 31.5–34.8% similarity with orthologous proteins of other filamentous fungi (31.5–34.8%) (Appendix A). Similarly, AoPdeL shares 93.3% and 72.7% sequence similarity with homologous protein from *D. flagrans* and *D. haptotyla*, respectively, and the sequence identity between AoPdeL and the orthologs of other filamentous fungi varied from 30.9–43.2% (Appendix A). Phylogenetic analysis showed that PdeH or PdeL orthologues from different fungi clustered in two clades, and orthologs from NT fungi were distributed in a single branch (clade II) (Appendix A).

### 2.2. AoPdeH and AoPdeL Regulate Mycelial Growth and Morphology

Three independent knockout strains for *AopdeH* (Δ*AopdeH*) and *AopdeL* (Δ*AopdeL*) were respectively confirmed (Appendix A) and used for phenotypic analysis. The growth rate and colony morphology of the WT and mutant strains were compared on different media. As a result, it was found that the Δ*AopdeH* mutants showed growth defects on the three media at 28 °C for five days post-induction (dpi), showing a significant defect in radial growth and reduction in aerial hyphae growth compared to the WT strain (Figure 1A,B). However, the mycelial growth of the Δ*AopdeL* mutant was similar to the WT strain. The number of hyphal septa had no significant difference between the WT and mutants, whereas partial hyphal cells of the Δ*AopdeH* and Δ*AopdeL* mutants became swollen (Appendix A). Moreover, the transcriptional levels of *AopdeH* and *AopdeL* in the WT strain were determined, and their expression increased with the extension of culture times; the expression of *AopdeH* and *AopdeL* were increased by 3.8-fold at 7 dpi compared to that at 3 dpi (Figure 1C).

### 2.3. AoPdeH and AoPdeL Regulate Intracellular cAMP Levels

The cAMP content in the mycelium of the WT and mutant strains was measured at different developmental stages. The results showed that the cAMP was in a dynamic state in the WT strain. The cAMP content increased gradually at the vegetative growth and early (12 h post-induction, hpi) and middle stage (24 hpi) of trap formation. In contrast, the cAMP content decreased in the stage of nematode predation (36 and 48 hpi). Compared to the WT strain, the cAMP content significantly increased (*p* < 0.05) in the mutants at the mycelial mature stage (seven days) and at stages of trap formation and nematode predation (Figure 1D). For example, in the mycelial mature stage (seven days), the mycelial cAMP content of the Δ*AopdeH* and Δ*AopdeL* mutants were 2.5-fold and 2.9-fold higher than the WT strain, respectively; In the stage of nematode predation (36 hpi), the mycelial cAMP content of the Δ*AopdeH* and Δ*AopdeL* mutants was 6.8-fold and 5.0-fold higher than the WT strain, respectively. These results showed that the absence of *AopdeH* and *AopdeL* resulted in an accumulation of intracellular cAMP levels.

### 2.4. AoPdeH and AoPdeL Affect Autophagy and Cellular Ultrastructure

The autophagic process in the WT and mutants was observed by staining with monodansylcadaverine (MDC) dye; many autophagosomes could be seen in the WT, which distribute in hyphal cells in a punctate pattern. In contrast, the number of autophagosomes was decreased in the hyphal cells of the Δ*AopdeH* and Δ*AopdeL* mutants, whereas the volume of autophagosomes in mutants became remarkably enlarged (Figure 2A). In addition, the cellular ultrastructures of the WT and mutants were observed using transmission electron microscopy (TEM; Hitachi, Tokyo, Japan). The vacuole was observed in the hyphal cells of the WT strain, and woronin body could be seen near the septum; similar to the WT, vacuole and woronin body were observed in the hyphal cells of the Δ*AopdeL* mutant, as well autophagosome-like structure could also be observed. In contrast, less woronin body was observed in the hyphal cells of the Δ*AopdeH* mutant, and more autophagosome-like structures were observed in the Δ*AopdeH* mutant (Figure 2B).

### 2.5. AoPdeH and AoPdeL Play Different Roles in Conidiation

To explore the regulatory role of PDEs in the asexual development of *A. oligospora*, we compared the asexual sporulation of the WT and mutants. Firstly, it was found by side-shooting that there was no significant difference between the conidia-forming method of the Δ*AopdeL* mutant strain and the WT (Figure 3A). Each conidiophore had a round of spores clustered by 4–5 spores. In contrast, the number of conidiophores of the Δ*AopdeH* mutant was significantly less than that of the WT, and there was almost no spore formation on the conidiophore (Figure 3A). In addition, the conidial morphology of the Δ*AopdeH* and Δ*AopdeL* mutants became slender, and their septa disappeared (Figure 3B). Further analysis showed that the conidia yield of the Δ*AopdeH* mutant was significantly reduced (only 4.2% of the WT), whereas the conidia yield of the Δ*AoPdeL* mutant was slightly less than the WT, and the difference was not significant (Figure 3C).

To further investigate the role of PDEs to conidiation, eight sporulation-related genes such as *abaA*, *aspB*, *flbC*, *fluG*, *velB*, *vosA*, *lreA*, and *lreB* were selected, and their transcriptional levels were determined during the early (three days), middle (five days), and later stages of sporulation. Compared with the WT, the transcription of *abaA* and *fluG* was significantly downregulated (*p* < 0.05) in the Δ*AopdeH* and Δ*AopdeL* mutants at different stages of sporulation. Additionally, the transcription of *aspB*, *velB*, *vosA*, *lreA*, and *lreB* was significantly downregulated (*p* < 0.05) in the Δ*AopdeH* mutant at the later stage of sporulation (Figure 3D), whereas the transcription of these genes had no change in the Δ*AopdeL* mutant at the same stage (Figure 3E).

### 2.6. AoPdeH and AoPdeL Play Different Roles in Stress Response

The WT and mutant stress response was compared against cell wall stress agents, oxidative and osmotic stresses to investigate the effects of PDEs on environmental stresses. The radial mycelial growth of the Δ*AopdeH* mutants was considerably inhibited by these stressors (Figure 4, Appendix A). The relative growth inhibition (RGI) value was calculated considering the influence of the growth defects of the mutant strain. Compared with the WT, The *AopdeH* mutants had a higher RGI value at H_2_O_2_ (2.5, 5.0, and 7.5 mM), menadione (0.03, 0.06, and 0.09 mM), sorbitol (0.25, 0.5, and 0.75 M), NaCl (0.2 M), SDS (0.35 and 0.52 mM), and Congo red (0.05 mM). In contrast, the *AopdeL*-deficient strain showed a WT phenotype to oxidants (H_2_O_2_ and menadione) (Figure 4), and it had a higher RGI value at 0.1 M NaCl (Appendix A). Nevertheless, the radial growth of the Δ*AopdeL* mutants was promoted at 0.08–0.12 mg/mL Congo red (Appendix A).

### 2.7. AoPdeH and AoPdeL Regulate Trap Formation and Nematicidal Activity

Trap formation was evaluated after the addition of nematodes. The WT strain produced immature traps containing one or two hyphal loops at 12 hpi, mature traps were observed, and nematodes were captured at 24 hpi. Then the majority of the nematodes were captured at 36 hpi and digested by the WT strain at 48 hpi (Figure 5A). However, the absence of *AopdeH* and *AopdeL* resulted in a remarkable reduction in trap formation. The Δ*AopdeH* mutant could not produce traps, and the number of traps produced by the Δ*AopdeL* mutant was considerably reduced compared with the WT (Figure 5A,B). For example, the WT strain produced approximately 309 and 811 traps per plate at 24 and 48 hpi, respectively, while the Δ*AopdeL* mutant produced 178 and 542 traps per plate at the same time points (Figure 5B). Correspondingly, the nematicidal efficiency of the Δ*AopdeH* mutant and Δ*AopdeL* mutant was remarkably decreased. At 24 hpi, the nematode mortality rates of WT, Δ*AopdeL*, and Δ*AopdeH* mutant strains were 40.1, 26.7, and 15.8%, respectively, while at 48 hpi, the nematode mortality rates of them were 97.6, 81.3, and 43%, respectively (Figure 5C).

### 2.8. AoPdeH Regulates Secondary Metabolism

The hyphae and supernatants of the WT and Δ*AopdeH* mutant strains were quantified, and the hyphal wet biomass of the WT (2.43 g) was 2.02-fold higher than that of the Δ*AopdeH* mutant (1.20 g). Based on the biomass levels of hyphae, 200 and 405 mL of supernatants were collected from the WT and Δ*AopdeH* mutant strains, respectively. The supernatants were extracted with ethyl acetate and vacuum-evaporated to obtain a crude extract. Then, the metabolites in the crude extracts were analyzed using liquid chromatography-mass spectrometry (LC-MS). According to the chromatogram, the WT strain produces plentiful metabolites, contrasting to the peaks of many compounds not detected or markedly decreased in the Δ*AopdeH* mutants (Appendix A). Volcano plot analysis showed substantially more downregulated compounds in the Δ*AopdeH* mutant than in the WT strain; there were 99 upregulated and 1855 downregulated compounds in the Δ*AopdeH* mutant compared with the WT (Figure 6A). Correspondingly, more downregulated metabolic pathways were found in the Δ*AopdeH* mutant (Figure 6B). The differential metabolic pathways were mainly enriched in degradation of the aromatic compounds toluene, naphthalene, mandelate, and L-tyrosine, biosynthesis of aromatic amino acid, 4-hydroxybenzoate, rosmarinic acid, novobiocin, and chorismate metabolism (Appendix A). In addition, arthrobotrisins, a kind of specific sesquiterpenyl epoxy-cyclohexenoid (SEC) metabolites found in *A. oligospora* and other NT fungi [29,30], were detected in the Δ*AopdeH* mutant and WT strain (diagnostic fragments ion at *m*/*z* 139, 393, and 429) (Figure 6C,D), and the relative peak area of arthrobotrisins in Δ*AopdeH* mutant (MA: 5319713) is remarkably less than that in the WT strain (MA: 76693347) (Appendix A).

## 3. Discussion

Previous studies have shown that PDEs can regulate intracellular cAMP levels in yeasts and filamentous fungi, thus playing a crucial regulatory role in nutrient sensing, asexual reproduction, and pathogenic development in yeasts and filamentous fungi [5,7,13,17,20]. In this study, we investigated the significance of PDE-mediated regulation of intracellular cAMP in the asexual and pathogenic development of *A. oligospora*. Our results showed that two PDEs AoPdeH and AoPdeL play a different role in the asexual growth, development, and pathogenicity of *A. oligospora*.

In *S. cerevisiae* and *C. albicans*, deletion of *pde2* resulted in increased accumulation of intracellular cAMP levels [7,31]. Similar results were observed in several filamentous fungi. In *M. oryzae*, the Δ*pdeL* mutant accumulated 1.5-fold higher levels of cAMP than the WT, while the Δ*pdeH* mutant and the Δ*pdeL*/Δ*pdeH* double mutant accumulated ~3-fold and ~4.5-fold higher levels of cAMP than the WT, respectively [11]. In *B. cinerea*, intracellular cAMP levels of the *bcpde1* deletion strains were similar to those of the WT, while Δ*bcpde2* and ΔΔ*ccpde1/2* mutants showed slightly decreased cAMP levels [13]. Similarly, the *A. flavus* Δ*pdeH* strain accumulated 1.8-fold higher cAMP levels than the WT and 1.6-fold higher than the Δ*pdeL* [5]. In this study, the absence of *AopdeH* and *AopdeL* caused the increased accumulation of intracellular cAMP levels during the hyphal stage, stages of trap formation, and nematode predation. Recently, resistance to inhibitors of cholinesterase 8 (Ric8), a negative regulator of G protein signaling, its deletion resulted in a significant reduction in cAMP level in *A. oligospora* [26]. In contrast, the deletion of genes encoding for regulators of G protein signaling (RGSs) caused increased intracellular cAMP levels in *A. oligospora* [25]. These findings suggested that PDEs govern G protein signaling by regulating cAMP levels in fungi.

As expected, the deletion of *AopdeH* caused significant growth defects. The radial growth of Δ*AopdeH* mutants became slow, and aerial hyphae were abnormally thin compared with the WT; however, there was no significant difference in the growth rate and colony morphology of the Δ*AopdeL* mutants compared to the WT strain. Similar growth defects caused by disruption of PDEs were found in several filamentous fungi such as *M. oryzae* [20], *B. cinerea* [13], and *A. flavus* [5]. In *M. oryzae*, the Δ*pdeH* colony was flat due to reduced aerial hyphal growth and displayed enhanced pigmentation and marginally slower radial growth, whereas the Δ*pdeL* was similar to the WT strain and showed no apparent defects in aerial or radial growth and colony morphology [20]. These findings showed that PdeH plays a crucial role in aerial or radial growth and colony morphology, but PdeL only plays a minor role in mycelial growth.

Compared with the WT strain, the conidiophores and conidia yield of the Δ*AopdeH* mutants were significantly reduced, and the conidial morphology was also deformed, becoming narrow and without a septum. In contrast, the conidiophores and conidia yield of Δ*AopdeL* mutants were not significantly different from that of WT. However, the Δ*AopdeL* spore morphology is similar to that of the Δ*AopdeH* mutants. Similarly, in *N. crassa*, the *pde2* deletion mutant does not produce any conidia [32]. In *B. cinerea*, deletion of *bcpde2* causes reduced spores (20%) compared to the WT strain, and the conidia morphology becomes narrow [13]. In contrast, in *M. oryzae*, the conidia yield of the Δ*pdeH* mutant is increased 2–3 folds. The number of conidiophores increased 2-fold, whereas the Δ*pdeL* mutant had no difference in conidia production or conidiophores from WT [20], but both Δ*pdeL* and Δ*pdeH* mutants produced elongated and thinner conidia [11]. In addition, the transcription of several sporulation-related genes such as *abaA*, *fluG*, and *lreB* was significantly downregulated (*p* < 0.05) in the Δ*AopdeH* mutant, which is consistent with the suppression of conidia yield. The AbaA and FluG are the key regulators of conidiation in several filamentous fungi, such as *Aspergillus nidulans* and *A. fumigatus* [33], and *Beauveria bassiana* [34,35]. Therefore, PdeH is indispensable for conidiation in *A. oligospora* and other fungi, and PdeL plays a specific role in conidial morphology.

Our previous studies revealed that G protein signaling regulates multi-stress responses in *A. oligospora* [26,36]. In this study, the absence of the *AopdeH* caused significant defects in tolerance to osmotic salts, oxidants, and cell wall stress agents; loss of *AopdeL* only altered the sensitivity to NaCl and Congo red. The results are similar to *S. cerevisiae*, deletion of the *pde2* causes the cells to be sensitive to freezing and thawing, and the oxidative stress response of the *pde2* deletion mutant can be induced by H_2_O_2_ or paraquat, but the absence of *pde1* does not affect cell stress resistance [7]. Similarly, the *Candida albicans pde2* deletion strains exhibited substantial growth defects in tolerance to ionic stress (NaCl and KCl) and high temperature, but not to sorbitol. However, *pde1* deletion did not affect the tolerance to these stresses [31]. In contrast, in *M. oryzae*, there are no phenotypic changes between the Δ*pdeH* mutant and the WT strain in response to stress agents, such as ionic stress, oxidative stress, osmotic stress, or cell wall-disturbing agents [11]. These findings showed that PDEs involve regulating stress signaling, and the function of PDEs is varied in different fungi, of which, PdeH likely plays a predominant role in regulating adverse stresses.

Previous studies have been revealed that PDEs are required for full virulence of *C. albicans* [31] and several pathogenic fungi [11,20]. In *C. albicans*, the adhesion ability of the Δ*pde2* mutant was approximately 10-fold lower than that of the WT, and deletion of *pde2* caused a drastic reduction in virulence and double deletion of *pde1* and *pde2* completely eliminated virulence [31]. In *M. oryzae*, loss of *pdeH* significantly accelerated appressorium formation on non-inductive surfaces. However, Δ*pdeH* conidia failed to infect the host efficiently and cause typical blast lesions owing to its compromised ability to form proper infection hyphae and further advance its growth and spread in the host tissue [11,20]. Similarly, deletion of *bcpde2* resulted in severely affected full virulence of *B. cinerea*, whereas the *bcpde1* deletion strain displayed a WT-like phenotype [13]. In this study, deletion of *AopdeH* abolished the trap formation and caused a remarkable reduction in predation efficiency, and deletion of *AopdeL* also hindered the trap formation and predation efficiency. These findings suggested that PDEs, especially PdeH, play a more prominent role than PdeL in infectious structure development and pathogenicity.

Like other filamentous fungi, NT fungi can produce numerous metabolites during vegetative growth and predator-prey interaction [30,37,38]. Recently, G protein signaling has been revealed to regulate the secondary metabolism in *A. oligospora*, such as the regulator of G-protein [26] and small GTPases [39]. In this study, the absence of *AopdeH* caused a remarkable alteration in secondary metabolism; 99 compounds were upregulated and 1855 downregulated in the Δ*AopdeH* mutant compared to the WT. In particular, the specific SEC metabolites arthrobotrisins were identified, the relative content of which decreased by 93.1% in the Δ*AopdeH* mutant. SEC metabolites can regulate the conidiation and morphological switch of NT fungi [30,38,40]. Similarly, the content of arthrobotrisins was downregulated, respectively, by 9.57- and 9.95-fold in the Δ*Aoras2* and Δ*Aorheb* mutants as compared with that in the WT strain [39]. Similarly, in *A. flavus*, deletion of both *pdeH* and *pdeL* caused a significant increase in aflatoxin production compared to the WT strain [5]. Therefore, PDEs play a vital role in secondary metabolism in NT fungi and other fungi.

PDEs are also involved in autophagy; loss of *AopdeH* and *AopdeL* caused a reduction in the number of autophagosomes and increased the volume of autophagosomes. Recently, several autophagy-related genes such as *atg1*, *atg4*, *atg5*, and *atg8* have been revealed to regulate conidiation and trap the formation of *A. oligospora* [41,42]. However, the regulatory mechanism of PDEs to the autophagic process needs to be studied further. In summary, our works revealed the important role of PDEs in the mycelial growth, autophagy, conidiation, stress response, trap formation, and secondary metabolism in *A. oligospora*, and AoPdeH plays a more prominent role than AoPdeL in regulating the growth, development, and pathogenicity through modulation of intracellular cAMP levels. Our results expanded the understanding of PDEs in NT fungi, which provides a basis for exploring the regulation mechanism underlying trap development and environmental adaptation of NT fungi.

## 4. Materials and Methods

### 4.1. Fungal Strains, Culture Conditions, and Media

The WT strain of *A. oligospora* (ATCC24927) and mutants generated in this study were routinely maintained as previously described [43]. Uracil-deficient *S. cerevisiae* FY834 was used as the host to construct the knockout vector, which was cultured in YPD (10 g/L yeast extract, 20 g/L peptone, and 20 g/L dextrose) [44]. *Escherichia coli* strain DH5α (Takara, Shiga, Japan) was used to store the plasmid pCSN44 containing the hygromycin resistance gene (*hph*) fragment and the plasmid pRS426 used to construct the knockout vector [45].

Potato dextrose agar (PDA), tryptone-glucose (TG), and tryptone yeast extract glucose agar (TYGA) media were prepared as previously described [43] and used to analyze mycelial growth. A liquid TG medium was used to prepare the mycelium for protoplast production or DNA extraction. TB3 medium (3 g/L hydrolyzed casein, 200 g/L sucrose, 3 g/L tryptone, and 0.75% agar powder) supplemented with 200 μg/mL hygromycin for protoplast regeneration and screening of positive transformants. Conidia were induced in the CMY medium (20 g/L maizena, 20 g/L agar, and 5 g/L yeast extract). Oatmeal water medium was used for culturing the nematode *Caenorhabditis elegans*, which was used for bioassay.

### 4.2. Sequence and Phylogenetic Analysis of PDEs

The orthologous AoPdeH (AOL_s00083g160) and AoPdeL (AOL_s00097g378) were identified in *A. oligospora* by comparing the protein sequences of homologous PDEs in the model fungi *S. cerevisiae*, *A. nidulans*, and *N. crassa*. The conserved domains of PDEs were predicted by InterProScan 4.8 (http://www.ebi.ac.uk/Tools/pfa/iprscan/, accessed on 19 October 2019), and the theoretical isoelectric point (pI) and molecular weight of PDEs were calculated using the pI/MW tool (http://web.expasy.org/compute_pi/, accessed on 19 October 2019). Orthologous PDEs from different fungi were downloaded from the GenBank, and a phylogenetic tree was constructed using the Mega software package (version 7.0) [46].

### 4.3. Vector Construction and Targeted Gene Deletion

*AopdeH* and *AopdeL* were disrupted by homologous recombination as described previously [44,47]. The hygromycin fragment *hph* was used as a selection marker conferring hygromycin B resistance on transformants. The homologous regions of the target gene and *hph* cassette were amplified by PCR using the primer pairs (Appendix A). The pRS426 plasmid was digested with *Eco*RI and *Xho*I, and the amplified fragments were co-transformed into *S. cerevisiae* by electroporation. The correct plasmid was screened on the SC-Ura plates and transformed into *E. coli* DH5α for expansion culture. The gene fragment for *AopdeH* and *AopdeL* disruption was amplified with primer pairs 160-5F/160-3R and 378-5F/378-3R (Appendix A) using the corresponding knockout vector plasmid as a template. Finally, the knockout construct was transformed into *A. oligospora* protoplasts by chemical transformation as previously described [47,48]. The hygromycin-resistant transformants were screened in TB3 plates and further confirmed by PCR amplification using primer pairs (Appendix A), respectively, and the transformants were further confirmed by Southern blot analysis as previously described [49].

### 4.4. Analysis of Mycelial Growth and Colonial Morphology

After synchronizing the WT and mutant strains, their growth rate and colonial morphology were compared on PDA, TYGA, and TG plates at 28 °C. The colonial diameters of the WT and mutant strains were determined every 24 h and photographed on the fifth day to record their colony morphology. The fungal cell wall and hyphal septum were dyed by Calcofluor white (Sigma-Aldrich, St. Louis, MO, USA) staining, as previously described [28], to observe hyphal morphology.

### 4.5. Analysis of Conidiation and Autophagy

The same size of the mycelial blocks was individually inoculated on the CMY medium and incubated at 28 °C for 14 days to evaluate the sporulation ability of the fungal strains. The conidiophore was observed using the side-shot approach described previously [50]. Conidia were harvested in 40 mL sterile water, followed by filtration through six layers of lens tissues to remove mycelial debris, then the conidial production was counted with a hemocytometer as previously described [25].

The WT and mutants were incubated on PDA plates with sterile coverslips at 28 °C for five days to detect autophagy. The mycelia were treated with 30–50 µL of 100 mg/mL MDC staining solution at 37 °C in the dark for 30 min. Then the images were observed by fluorescence microscopy (Leica, Mannheim, Germany).

### 4.6. Quantification of Intracellular cAMP

Quantification of intracellular levels of cAMP was carried out as previously described [51]. The WT and mutant strains were incubated on TYGA plates at 28 °C, and mycelial samples were harvested at 3 and 7 dpi, respectively. In addition, the strains were incubated on TYGA plates for seven days, then about 300 nematodes were added to the plates, and mycelia were harvested at 12, 24, 36, and 48 hpi. The mycelia samples were treated with 1 M HCl for 30 min, then frozen in liquid nitrogen. Intracellular cAMP was extracted, and cAMP levels were quantified using the Direct cAMP ELISA kit (Enzo Life Sciences, Farmingdale, NY, USA).

### 4.7. Analysis of Stress Tolerance

Several chemical stressors, such as cell wall stress agents (SDS and Congo red), oxidants (menadione and H_2_O_2_), and osmotic agents (NaCl and sorbitol), were used to evaluate the stress response of the WT and mutants as previously described [25,27]. The strains were cultured at 28 °C for five days, the diameter of the colony was measured and photographed to record the colony morphology. The RGI value of the mutant versus WT strain was calculated as previously described [25,27].

### 4.8. Trap Formation and Bioassay

Approximately 3 × 10^4^ conidia of WT, Δ*AopdeH*, and Δ*AopdeL* mutant strains were inoculated in water agar plates, after culturing at 28 °C for five days, and about 300 nematodes were added to the plates to analyze the trap formation and nematode predation efficiency, respectively. The number of traps and captured nematodes were calculated at 24 and 48 hpi, respectively.

### 4.9. Reverse Transcription-Quantitative PCR (RT-qPCR)

The fungal strains were cultured in TYGA plates at 28 °C, and mycelial samples were collected at 3, 5, and 7 dpi. Total RNA of mycelia was isolated and used to reverse-transcribe the cDNA. The cDNA from each strain was used as a template for RT-qPCR, and the *A. oligospora* β-tubulin gene was used as an internal standard (Appendix A). The transcriptional levels of *AopdeH*, *AopdeL*, and sporulation-related genes at different stages were determined by RT-qPCR. The relative transcript levels of each gene were calculated using the 2^−ΔΔCt^ method [52].

### 4.10. Comparison of Metabolites between WT and ΔAopdeH Mutant Strains

The WT and Δ*AopdeH* mutant strains were cultured for seven days in PD broth at 28 °C and 180 rpm. The hyphae of fermentation broth of the WT and Δ*AopdeH* mutant strains were harvested and qualified. The fermentation broth was treated, and the metabolites were analyzed with LC-MS as previously described [39]. Three replicates were maintained for the analysis. Untargeted metabolomics analysis was performed using the Compound Discoverer 3.0 software (Thermo Fisher Scientific, Waltham, MA, USA).

### 4.11. Statistical Analyses

Data were expressed as standard deviations (SDs) of the means from three biological replicates, and subjected to one-way analysis of variance, followed by Tukey’s honestly significant difference (HSD) test. *p* < 0.05 indicates a significant difference.

## Figures and Tables

**Figure 1 pathogens-11-00405-f001:**
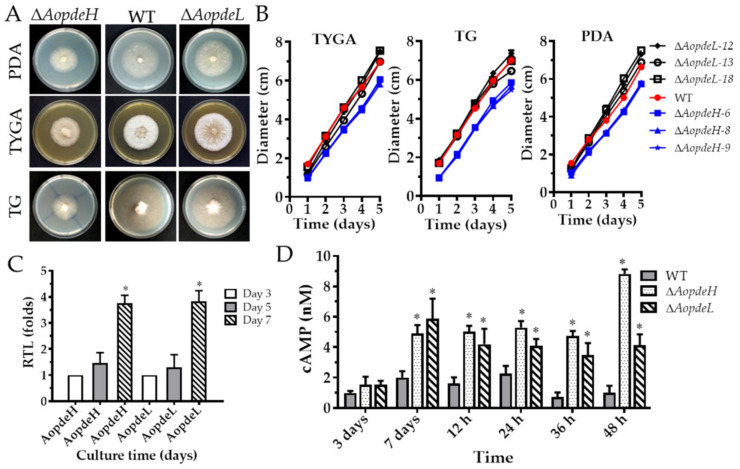
Comparison of mycelial growth, transcript of PDEs genes, and cAMP levels between WT and mutants. (**A**) Colonial morphologies of WT and mutants incubated on different media at 28 °C for five days. (**B**) Colonial diameters of WT and mutants incubated on different media. (**C**) Relative transcription levels (RTLs) of *AopdeH* and *AopdeL* in WT strain at different culture times. An asterisk indicates a significant difference between the RTL of *AopdeH* and *AopdeL* on the seventh day and the third day (Tukey’s HSD, *p* < 0.05). (**D**) Comparison of cAMP levels in the WT and mutant at different culture times and trap induction. An asterisk indicates a significant difference between Δ*AopdeH* and Δ*AopdeL* mutants and WT strain (Tukey’s HSD, *p* < 0.05).

**Figure 2 pathogens-11-00405-f002:**
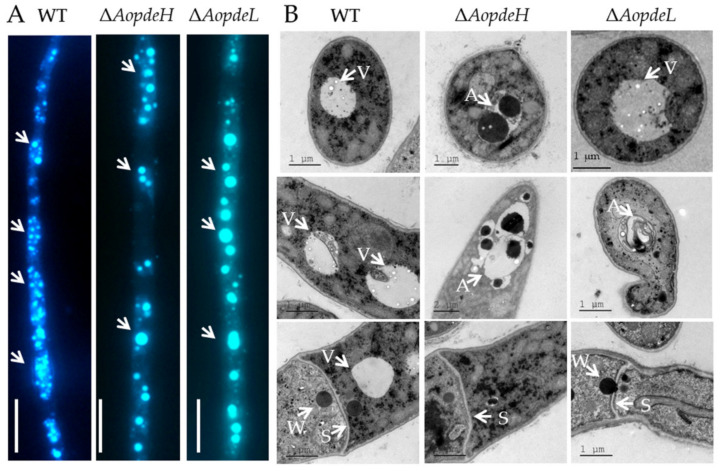
Comparison of autophagy and cellular ultrastructure between WT and mutants. (**A**) Comparison of autophagosomes in the hyphal cells of the WT and mutants. White arrows: autophagosomes. Bar = 10 μm. (**B**) Cellular ultrastructure of the WT and mutants. V, vacuole; S, hyphal septum; W, woronin body; A, autophagic vacuole.

**Figure 3 pathogens-11-00405-f003:**
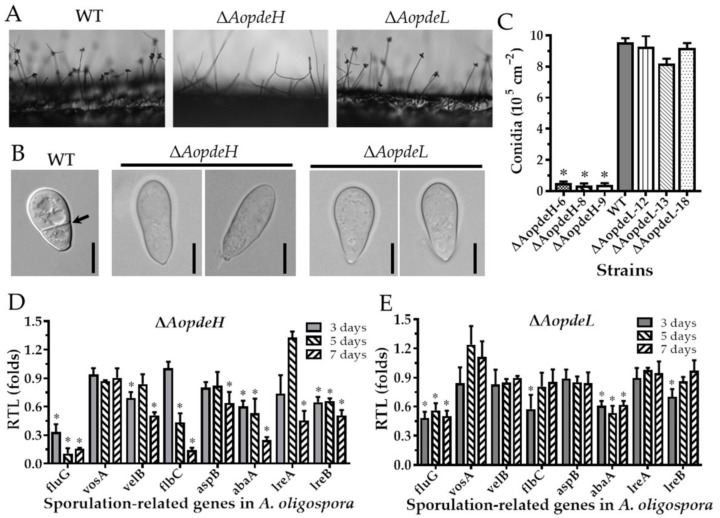
Comparison of conidiation, spore morphology, and transcript of sporulation-related genes between WT and mutants. (**A**) Conidiophores of the WT and mutants. (**B**) Spore morphology of the WT and mutants. (**C**) Conidia yields of the WT and mutants. (**D**,**E**) Relative transcription levels (RTLs) of sporulation-related genes in Δ*AopdeH* and Δ*AopdeL* mutant compared to WT strain at different culture times. An asterisk (**C**–**E**) indicates a significant difference between mutant and WT strains (Tukey’s HSD, *p* < 0.05).

**Figure 4 pathogens-11-00405-f004:**
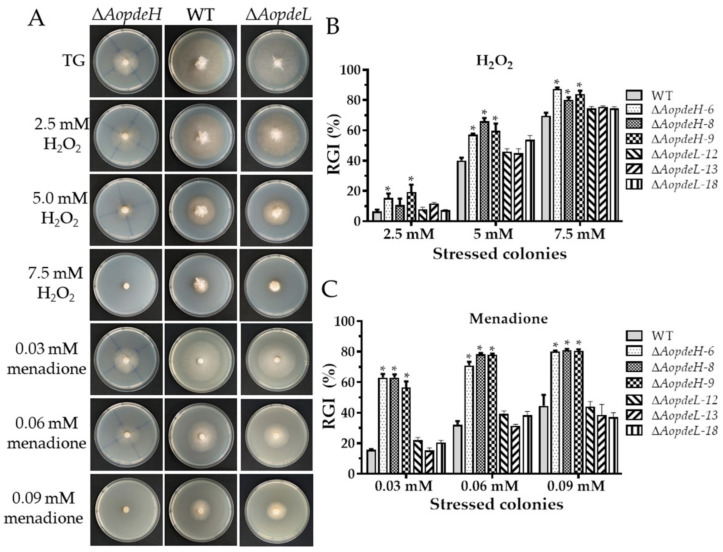
Comparison of oxidative stress responses between WT and mutants. (**A**) Colonial morphology of fungal strains under oxidative stresses. (**B**,**C**) Relative growth inhibition (RGI) of fungal colonies after being grown for five days at 28 °C on TG plates supplemented with different concentrations of H_2_O_2_ and menadione, respectively. An asterisk indicates a significant difference between mutant and the WT strain (Tukey’s HSD, *p* < 0.05).

**Figure 5 pathogens-11-00405-f005:**
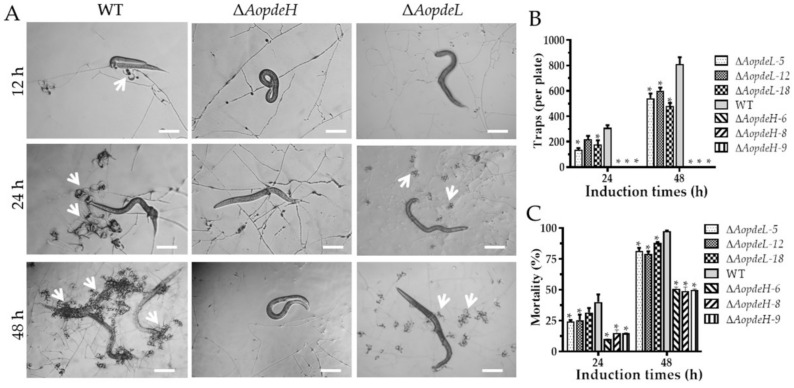
Comparison of trap formation and nematode predation efficiency between WT and mutants. (**A**) Trap formation and nematode predation at different time points. White arrows: Traps. Bar = 100 μm. (**B**) Comparison of traps produced by WT and mutants at 24 and 48 h. (**C**) Comparison of nematodes captured by WT and mutants at 24 and 48 h. An asterisk (**B**,**C**) indicates a significant difference between mutant and the WT strain (Tukey’s HSD, *p* < 0.05).

**Figure 6 pathogens-11-00405-f006:**
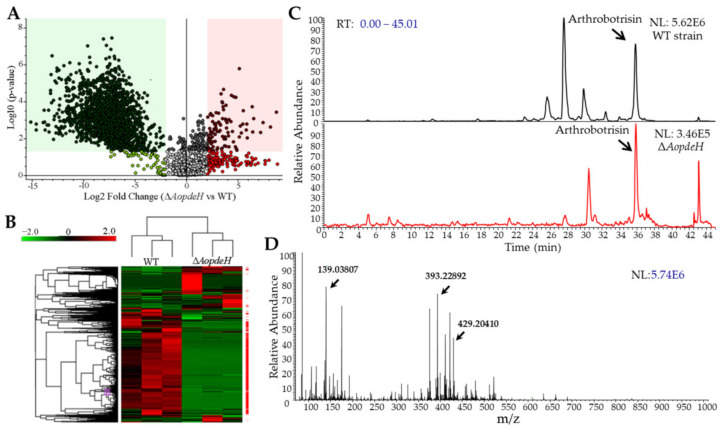
Comparison of differential expressed metabolites between WT and Δ*AopdeH* mutant strains. (**A**) Volcano plot of differential metabolites between Δ*AopdeH* mutant and WT strain. (**B**) Heatmap of differential metabolic pathways between Δ*AopdeH* mutant and WT strain determined via KEGG enrichment. (**C**) Comparison of ion chromatograms of arthrobotrisin between WT and mutant strains. Arrow, the peak of arthrobotrisin. (**D**) Mass spectrogram of arthrobotrisins in the WT strain. Arrow, diagnostic fragments ion at *m*/*z* 139, 393, and 429.

## Data Availability

Not applicable.

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
