# Peer review of "Functional Analysis of Two Affinity cAMP Phosphodiesterases in the Nematode-Trapping Fungus Arthrobotrys oligospora"

_pathogens, 2022, doi:10.3390/pathogens11040405_

Round 1

Reviewer 1 Report

Results

I suggest the authors to thoroughly check the table legends and the quality of the table and figures; particularly I invite the authors to carefully revise the figura1 D.

Statistical analyses

“the variance between the WT and mutant was ranked by Tukey’s honestly  significant difference (HSD) test. p < 0.05 indicates a significant difference”: It is not the variances that are compared in a Tukey test.

I recommend, that the data in percentage be transformed

Author Response

Results

I suggest the authors to thoroughly check the table legends and the quality of the table and figures; particularly I invite the authors to carefully revise the figura1 D.

Response: Thank you for your kindly suggestion. We have thoroughly checked the table legends and the quality of the table and figures, and revised the manuscript accordingly.

Statistical analyses

“the variance between the WT and mutant was ranked by Tukey’s honestly  significant difference (HSD) test. p < 0.05 indicates a significant difference”: It is not the variances that are compared in a Tukey test.

Response: Thank you. We have revised the sentence.

I recommend, that the data in percentage be transformed

Response: Done. The data in percentage have been transformed accordingly.

Reviewer 2 Report

It is original and relevant work using a fungus that is a biological controller. The work is interesting as it elucidates sero proteases from the fungus Arthrobotrys oligospora. I agree with the conclusion of the paper. This work expands the roles of Phosphodiesterases and deepens the understanding of the regulation of trap formation in nematode-trapping fungi.

The article is very well written, just watch out for some names in italics as in lines 90 and 93.

Author Response

Thank you for your positive comments. We have checked the sentence in lines 90 and 93. The AoPdeH and AoPdeL indicate the names of proteins, so they do not appear in italics.

Reviewer 3 Report

The authors present an experimental study on the effects of phosphodiesterases via G-proteins on various phenotypes. The fungus in question, Arthrobotrys sp. has, in this respect, not been analysed in depth before. As the data appear sound, and setup as well as structure of the contribution is reasonable, the reviewer sees no reason not to accept the manuscript for publication after minor revision. 

The reviewer appreciates the thorough investigation on autophagy and - especially - on conidiation. Of course, with respect to the special interest in Arthrobotrys, trap formation is probably the most important issue.

The effects of the AoPde mutants has been studied with respect to quantitiy of traps and the potential to capture nematodes. The reviewer wonders, if there have been no effects on morphology and maybe the time course of trap formation. The authors should discuss this interesting question and maybe share their observations.

It would also help to improve the impact of the contribution to discuss in more depth the effects on conidiation of the genes in relation to other ascomycetes.

One additional little thing: Sometimes non-mycologist readers would probably appreciate to read non-abbreviated genus names. At least at the beginning of paragraphs, the full name should always be given. The authors should also check the necessity for other abbreviations. These always render reading more difficult and can often easily be avoided. Especially abstracts should be completely free of abbreviations. There is no need for inserting the abbreviations PDE or WT directly after naming the complete terminus. 

On the whole, the authors submit an important and useful contribution that can be accepted after minor but careful revision. 

Author Response

Thank you for your positive comments. We are very appreciative of your great help to improve our manuscript. We have carefully considered and revised the manuscript according to your comments.